# Benchmark for Compositional Text-to-Image Synthesis

**Dong Huk Park**\*, **Samaneh Azadi**,\* **Xihui Liu, Trevor Darrell, Anna Rohrbach**
University of California, Berkeley

## Abstract

Rapid progress in text-to-image generation has been often measured by Frećhet Inception Distance (FID) to capture how realistic the generated images are, or by R-Precision to assess if they are well conditioned on the given textual descriptions. However, a systematic study on how well the text-to-image synthesis models generalize to novel word compositions is missing. In this work, we focus on assessing how true the generated images are to the input texts in this particularly challenging scenario of novel compositions. We present the first systematic study of text-to-image generation on zero-shot compositional splits targeting two scenarios, unseen object-color (*e.g.* "blue petal") and object-shape (*e.g.* "long beak") phrases. We create new benchmarks building on the existing CUB and Oxford Flowers datasets. We also propose a new metric, based on a powerful vision-and-language CLIP model, which we leverage to compute R-Precision. This is in contrast to the common approach where the same retrieval model is used during training and evaluation, potentially leading to biased behavior. We experiment with several recent text-to-image generation methods. Our automatic and human evaluation confirm that there is indeed a gap in performance when encountering previously unseen phrases. We show that the image *correctness* rather than purely *perceptual quality* is especially impacted. Finally, our CLIP-R-Precision metric demonstrates better correlation with human judgments than the commonly used metric. Dataset and evaluation code at: `https://github.com/Seth-Park/comp-t2i-dataset`.

## 1 Introduction

Text-to-image synthesis, which aims at generating an image based on an input textual description, has made large advances over the last few years [4, 37, 55]. While most text-to-image synthesis methods focus on improving the perceptual quality of synthesized images, it remains unclear whether these models have the ability to synthesize *novel* images or whether they simply *e.g.* memorize the training set. Compositionality is a key feature of visual intelligence, and it has been explored in visual perception [28] and image captioning [31] before. However, to the best of our knowledge, it has never been systematically studied for text-to-image synthesis. Humans who previously saw concepts "blue" and "flower", can easily imagine "a blue flower" even if they have not seen that specific combination before. But can the text-to-image synthesis models generalize to such novel compositions of concepts? Figure 1 showcases two examples, where the model fails to synthesize novel compositions such as "purple bill" and "black tips" correctly. Moreover, we also often observe some unintended changes of the background, pose, and other attributes when introducing the novel compositions, indicating some amount of entanglement between all these characteristics.

In this work, we provide the first systematic study of generalization to novel compositions for state-of-the-art text-to-image generation methods. We observe that there are relatively few novel compositions in the conventional test splits of commonly used text-to-image synthesis datasets, and no existing splits specifically targeting this scenario. Thus, we propose new benchmarks for evaluating compositional text-to-image synthesis, based on the Caltech-UCSD Birds (CUB) dataset [54] and Oxford-102

---

\*The first two authors contributed equally to this work.

35th Conference on Neural Information Processing Systems (NeurIPS 2021) Track on Datasets and Benchmarks.

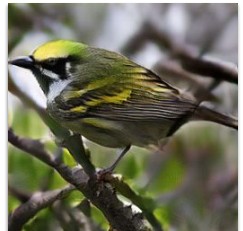
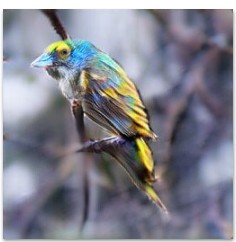
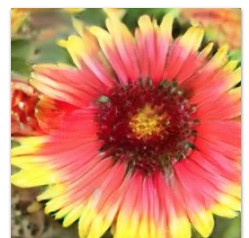
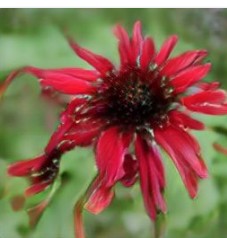

the bird has a **black bill**, **yellow crown**, and **yellow wingbar**.

the bird has a **purple bill**, **yellow crown**, and **orange wingbar**.

this flower has **petals** that are **red** and has **yellow tips**.

this flower has **petals** that are **red** and has **black tips**.

Figure 1: Seen adjective-noun pairs (illustrated in green) are swapped with unseen adjective-noun pairs (illustrated in red). All images are synthesized by DMGAN [63] given a caption and a fixed noise vector as input. The swapped attributes are not accurately depicted in the output images.

(Flowers) dataset [32], augmented with human-generated captions [40], which are commonly used in the text-to-image synthesis literature. Concretely, in both cases, the test set is composed of *Seen* image-caption pairs (with adjective-noun pairs that have appeared in the training set), *Unseen* pairs (with adjective-noun pairs that have not appeared in the training set), and *Swapped* pairs (generated from the Seen pairs by swapping seen adjective-noun pairs with unseen ones). Motivated by recent work which shows that some object properties (*e.g.* texture) are "easier" to learn than others (*e.g.* shape) [7], we propose two compositional scenarios, one targeting object-color compositions (*e.g.* "blue petal"), another targeting novel object-shape compositions (*e.g.* "long beak"). We denote the proposed compositional benchmarks as C-CUB Color/Shape and C-Flowers Color/Shape.

There are two types of evaluation metrics typically used in text-to-image synthesis, some measure image fidelity (perceptual quality) and others measure image correctness. Fréchet Inception Distance (FID) [9] is commonly adopted for evaluating the quality and diversity of the synthesized images. For evaluating the correctness of the synthesized images, previous works use R-Precision which measures whether based on the generated image we can retrieve the original description from a set of distractors. One caveat is that the text-to-image retrieval model used for computing the R-Precision is often the same model as used for training the text-to-image synthesis network. As a result, model bias is inevitably introduced during training. To mitigate this problem, we propose a new evaluation metric, named CLIP-R-Precision, which performs text-to-image retrieval based on the large-scale pretrained multimodal CLIP model [38]. Our human evaluation indicates that our proposed metric is more consistent with human judgments than the metric used in prior work.

Finally, we benchmark three representative recent text-to-image synthesis methods, DM-GAN [63], ControlGAN [19], and DF-GAN [48], on our proposed compositional data splits and CLIP-R-Precision evaluation metric. We perform both automatic and human evaluations. Our results demonstrate that the novel compositions indeed lead to a drop in performance. In fact, there is higher degradation in image *correctness* than image *perceptual quality*. We provide a qualitative analysis, shedding light onto feature entanglement as a possible cause of poor compositional generalization.

To sum up, in this work we establish the first benchmark for evaluating and analyzing how well text-to-image synthesis methods generalize to novel compositions of concepts. Our contributions are three-fold: We create new data splits with the Unseen and Swapped test samples, which are specifically designed for evaluating and analyzing how well the model can generalize to unseen compositions of concepts, such as object-color and object-shape. We propose a new evaluation metric for text-to-image synthesis, named CLIP-R-Precision, which better correlates with human judgement and overcomes the model bias introduced during training for the commonly used R-Precision. We benchmark and analyze three representative text-to-image synthesis methods with our proposed data splits and evaluation metrics, and demonstrate performance degradation in all unseen scenarios.

## 2 Related Work

**Text-to-image synthesis** Text-to-image synthesis has made great progress in the recent years, in particular leveraging generative adversarial networks (GANs) [8]. GAN-INT-CLS [41] and GAWWN [42] were the first attempts on GAN-based text-to-image synthesis. StackGAN [60] and several others [61, 62, 6, 15] improved the quality of synthesized images by using stacked generator

structures to sythesize images in a coarse-to-fine scheme. A prominent method, AttnGAN [55], incorporated attention-driven, multi-stage refinement for fine-grained text-to-image generation; it became a basis for many future works [13]. DM-GAN [63] introduced dynamic memory to progressively refine the generated images. ControlGAN [19] introduced word-level spatial and channel attention for the generator, along with the word-level discriminator and perceptual loss. DF-GAN [48] introduced a one-stage text-to-image backbone with a novel fusion module for the generator and a target-aware discriminator. Other approaches explored introducing memory mechanisms [63, 24], cycle-consistency constraint [37, 17], siamese networks [57, 47], and prior knowledge [36, 4] to improve the text-to-image synthesis performance. For synthesizing images with complex scenes, some works decompose the scene into multiple objects, and use the object layout or semantic layout as a bridge to synthesize the whole image [11, 21, 10], or generate different objects sequentially [46]. Scene graph-to-image synthesis [16] is related to this line of work, as they both consider an image as a composition of multiple objects (and object relationships). However, as opposed to the natural language description inputs, the structured information and the layout for image synthesis is directly provided by the scene graph input. Recent works explored a transformer-based vision-language representation model [5], cross-modal constrastive learning [59], and transformer and discrete VAE [39] for text-to-image synthesis.

**Evaluation metrics** Most of the prior attempts on text-to-image synthesis are evaluated using Inception Score (IS) [45] and Fréchet Inception Distance (FID) [9] for image fidelity. Precision and recall [44, 18] are alternative evaluation metrics to evaluate image quality and diversity for unconditional and conditional image synthesis. For text-to-image synthesis evaluation, R-precision is adopted to evaluate how well the generated image aligns with the text. In this work, we analyze the drawbacks of the previous evaluation metrics, and propose our new benchmark and evaluation metric to measure the compositionality of text-to-image synthesis models. We experiment with DM-GAN [63], ControlGAN [19] and DF-GAN [48], three representative approaches discussed above.

**Compositionality in visual systems** Compositionality is one of key properties of intelligent visual systems. Exploring the compositionality of visual systems is important for generalization of neural networks, few-shot learning, and overcoming dataset bias. A typical example is: given images of a "young tiger" and an "old car", is the model able to recognize an "old tiger"? In visual recognition, researchers explored the compositionality of objects and attributes so that the model can generalize to unseen combinations of visual concepts [28, 29, 53, 50, 35, 30, 23, 56, 26]. Compositionality has also been explored in visual relationship detection [34, 1] and action recognition [27]. Most related to our work is compositionality in image captioning and image synthesis. Previous work has explored the ability of generating novel image captions for unseen composition of concepts [31]. For image synthesis, a few works aim at object-level compositionality such as adding a pair of sunglasses to a face [51, 3, 58]. Another line of work explores disentanglement of different factors for image synthesis [2, 25, 14, 43]. MixNMatch [22] disentangled multiple factors including background, pose, shape and texture by decomposing image synthesis into multiple steps. However, the compositionality for text-to-image synthesis has never been explored in previous works. Here, we establish a new benchmark for compositional text-to-image synthesis and analyze the compositionality of several existing text-to-image synthesis models.

## 3 Constructing Compositional Splits for Text-to-Image Synthesis

In this section, we introduce a task that requires compositional understanding of adjective and object concepts when synthesizing images from input texts. To systematically analyze the performance of text-to-image models in generalizing to unseen composition of concepts, we propose multiple new compositional splits built from two datasets commonly used for text-to-image synthesis.

### 3.1 Problem Definition

Here, we define the compositional text-to-image synthesis task in which the goal is to measure how well a model generalizes to text inputs that contain unseen combinations of concepts. We assume a dataset $D$ containing $N$ images each described by $K$ captions: $D = \{(x^1, s^1_1, \ldots, s^1_K), \ldots, (x^N, s^N_1, \ldots, s^N_K)\}$. Within each caption, we identify a set of concept pairs,

Table 1: Dataset statistics for C-CUB (left) and C-Flowers (right) datasets. Num Imgs, Num Caps, and Avg Cap Len denote the total number of images, total number of captions, and average caption length, respectively. The number in parentheses is the number of heldout adjective-noun pairs.

| Type | Split | Num Imgs | Num Caps | Avg Cap Len |
|------|-------|----------|----------|-------------|
| Color (29) | Train | 9,010 | 84,709 | 15.21 |
| | Test Seen | 1,389 | 13,022 | 15.28 |
| | Test Unseen | 1,389 | 13,074 | 15.59 |
| | Test Swapped | 1,389 | 8,991 | 15.73 |
| Shape (34) | Train | 8,918 | 83,806 | 15.25 |
| | Test Seen | 1,435 | 13,451 | 15.21 |
| | Test Unseen | 1,435 | 13,548 | 15.40 |
| | Test Swapped | 1,435 | 6,208 | 15.70 |

| Type | Split | Num Imgs | Num Caps | Avg Cap Len |
|------|-------|----------|----------|-------------|
| Color (18) | Train | 6,637 | 59,111 | 13.36 |
| | Test Seen | 776 | 6,919 | 13.40 |
| | Test Unseen | 776 | 6,927 | 13.69 |
| | Test Swapped | 776 | 6,789 | 13.38 |
| Shape (17) | Train | 6,458 | 22,235 | 13.88 |
| | Test Seen | 718 | 2,558 | 13.86 |
| | Test Unseen | 718 | 3,045 | 13.79 |
| | Test Swapped | 718 | 2,463 | 13.83 |

specifically we consider adjective-noun pairs in this work. Although each individual concept, *i.e.* an adjective or a noun, is observed during training, some concept compositions are held out from the training set. For instance, the training captions may contain individual colors (*e.g. black, purple*), individual body parts (*e.g. bill*), and a subset of their compositions (*e.g. black bill*) while missing some other compositions (*e.g. purple bill*). Images and captions containing the held-out compositions are reserved for evaluation. We design three different test splits to measure (1) how well the model performs on the seen adjective-noun combinations observed during training (Test *Seen*), (2) how well the model generalizes to the unseen concept pairs reserved from training (Test *Unseen*), and (3) how well the model handles input texts in which seen concept pairs are swapped to become unseen (Test *Swapped*). The motivation for the third split is to prevent any other factors besides novel adjective-noun compositions from affecting the model's generalization performance. To achieve this, we introduce minimal changes to *Seen* split by *swapping* adjectives to form *unseen* compositions. For instance, a test seen caption is "This is a bird with red beak which is curvy and thick" where "red beak" is a seen composition. While a test unseen caption is "This is a magnificent bird with brown crown, striking yellow cape, orange upper wings, and blue beak". If the quality of the synthesized image from the unseen caption is not good, it is hard to tell whether it is because of the unseen composition (blue beak) or because the input caption is longer and more complicated. In order to control the various factors and analyze the effect of novel compositions, we introduce the test swapped caption, "This is a bird with blue beak which is curvy and thick" where it adopts the identical sentence structure and vocabularies with the seen caption, and only the unseen combination of "blue beak" is introduced.

## 3.2 Compositional Split Generation

In order to create the aforementioned compositional splits, we first identify representative sets of nouns and adjectives present in the dataset, and curate a list of synonyms for each word to account for the variations in how they manifest in the captions. When selecting adjectives, we first determine the 60 most frequent adjectives that appear in the dataset. Then we filter out ones that are either color-related or shape-related. Then, we find 100 most frequent adjective-noun pairs that are associated with the selected adjectives and extract all the nouns. We use Spacy [12] to tag, lemmatize, and parse the captions.

Once the nouns and adjectives are determined, we select the "novel" adjective-noun pairs that will constitute the evaluation set. Specifically, we calculate the frequencies of all the adjective-noun pairs and sort them from the most to the least frequent. Then, 10% of the unique adjective-noun pairs are withheld to become *unseen*; they are randomly sampled from between the 25th and 75th percentiles of the sorted list. (Sampling heldout pairs from the top of the list, *i.e.* most frequent ones, results in significant shrinkage of the training dataset and limits the variations in nouns present in the evaluation splits as there exists certain nouns that appear more frequently across pairs, while sampling from the long-tail results in a small test set.) Finally, based on the withheld pairs the dataset is split into $D_{train}$, $D_{test\ seen}$, and $D_{test\ unseen}$. When generating $D_{test\ swapped}$ from $D_{test\ seen}$, we keep the nouns and only modify the adjectives so that they form unseen pairs. Given the limited number of adjectives available in the heldout set, such swapping process can lead to certain heldout pairs dominating the split. To address this, we identify these dominant pairs and try to avoid introducing them if there are other candidates in the caption that can be swapped. Even with such measure, the frequencies of

heldout pairs can still be imbalanced, so during the computation of automatic metrics, we make sure that the dominant pairs do not overwhelm the score computation.

Following this procedure, we construct four different benchmarks from two commonly used datasets for text-to-image synthesis: Caltech-UCSD Birds (CUB) [52], a collection of 200 bird species, and Oxford-102 (Flowers) [33] which consists of 102 flower categories. The captions for both datasets are obtained from [40], who collected 10 human descriptions for each image. The license details of each dataset are in the respective papers. We denote the four different benchmarks as C-CUB Color, C-CUB Shape, C-Flowers Color, C-Flowers Shape, and their respective statistics can be found in Table 1. For more details on the proposed benchmarks, please refer to the Supplemental.

## 4   Evaluation for Text-to-Image Synthesis

Here, we review the commonly used evaluation metrics, discuss their limitations, and propose a new metric, CLIP-R-Precision, for measuring generated image correctness for the given input description.

**Fréchet Inception Distance**. A common metric to evaluate the performance of GANs in terms of quality and diversity of the synthesized samples specially in unconditional image synthesis is Fréchet Inception Distance (FID) [9]. It calculates the Fréchet distance between the embedding of real images extracted from a pre-trained Inception network and that of synthesized images. In the conditional text-to-image synthesis problem, while this metric evaluates the perceptual quality of the generated samples, it does not capture whether these images are well conditioned on the input text. Instead, R-Precision is used to evaluate the correctness of the generated images with respect to the given caption. In the following, we review R-Precision and shed some lights on its limitations, which we address with our new proposed metric, CLIP-R-Precision, discussed thereafter.

**R-Precision**. R-Precision calculates the top-R retrieval accuracy when retrieving the matching text from 100 text candidates using the generated image as a query. Typically $R = 1$, meaning that we calculate the top-1 retrieval accuracy. The text-image similarity score for retrieval is calculated using the Deep Attentional Multimodal Similarity Model (DAMSM) [55] consisting of an image encoder and a text encoder to map each sub-region of an image and its corresponding word in the sentence to a joint embedding space. This model is pre-trained on the real image-text pairs from the training data, and measures the fine-grained image-text similarity for retrieval. While the purpose of R-Precision is to measure how well the generated images align with the text inputs, we find that majority of the text-to-image synthesis models directly optimize the DAMSM module used in computing R-Precision [63, 19]. This results in text-to-image generation systems that are optimized specifically for DAMSM-based evaluation metric, making the R-precision metric model-specific. The issue with such a metric is that models that are not specifically tuned on DAMSM modules will likely perform worse on the metric, thereby making the metric less objective and necessitating model designs to include DAMSM modules which may not always be desirable.

**CLIP-R-Precision**. To address this, we propose a new evaluation metric, named CLIP-R-Precision. We adopt the recent multimodal CLIP model [38] instead of the standard DAMSM to calculate the R-Precision scores. CLIP is trained on a large corpus on Web-based image-caption pairs. It learns to bring the two embeddings (visual and textual) together via a contrastive objective. We finetune the pretrained CLIP model on the entire CUB and Oxford-102 dataset, respectively. Compared to DAMSM, CLIP obtains higher image-text retrieval performance on the standard test splits of the CUB and Flowers datasets: 26.20 vs. 22.1 and 20.0 vs. 18.44, respectively (with real images). Overall, our proposed CLIP-R-Precision is a stronger model-agnostic (i.e. disassociated in model training) evaluation metric compared to the standard R-Precision. We further validate this claim by analyzing correlation between human judgments, regular R-Precision and CLIP-R-Precision in Section 5.2.

## 5   Experiments

In this section, we study the performance of the existing text-to-image synthesis models in generalizing to the unseen composition of concepts that are introduced by our compositional data splits.

**Text-to-Image Synthesis Models.** We benchmark three text-to-image synthesis methods. DMGAN (Dynamic Memory GAN) [63] first generates an initial image and refines it iteratively with a Dynamic Memory based Image Refinement module. ControlGAN[19] aims at the user-controllable text-to-

Table 2: Benchmarking on the C-CUB Color and C-CUB Shape datasets.

| Type | Model | Split | FID↓ | R-Precicion (R=1)↑ | CLIP R-Precision (R=1)↑ |
|------|-------|-------|------|---------------------|--------------------------|
| Color | DMGAN | Seen | 14.38 | 64.59 ± 1.33 | 53.52 ± 0.93 |
| | | Unseen | 15.92 | 58.82 ± 1.83 | 50.71 ± 0.84 |
| | | Swapped | 14.31 | 56.22 ± 1.47 | 42.10 ± 0.78 |
| | ControlGAN | Seen | 16.58 | 56.35 ± 1.30 | 39.48 ± 0.68 |
| | | Unseen | 19.65 | 51.00 ± 1.38 | 36.71 ± 1.13 |
| | | Swapped | 16.60 | 48.26 ± 0.80 | 29.84 ± 1.65 |
| | DF-GAN | Seen | 16.61 | 32.83 ± 1.03 | 29.12 ± 0.78 |
| | | Unseen | 17.12 | 30.50 ± 1.65 | 27.74 ± 1.22 |
| | | Swapped | 17.19 | 29.34 ± 2.73 | 21.76 ± 1.16 |
| Shape | DMGAN | Seen | 13.34 | 65.52 ± 1.52 | 56.77 ± 1.48 |
| | | Unseen | 13.26 | 60.90 ± 1.35 | 49.89 ± 1.06 |
| | | Swapped | 15.15 | 58.69 ± 2.90 | 50.00 ± 2.20 |
| | ControlGAN | Seen | 15.80 | 56.07 ± 1.21 | 38.08 ± 0.85 |
| | | Unseen | 16.94 | 55.63 ± 1.24 | 34.69 ± 1.22 |
| | | Swapped | 16.76 | 53.57 ± 2.56 | 36.68 ± 2.42 |
| | DF-GAN | Seen | 17.59 | 28.67 ± 1.13 | 26.32 ± 0.78 |
| | | Unseen | 19.71 | 27.82 ± 0.80 | 23.84 ± 1.02 |
| | | Swapped | 19.99 | 25.56 ± 1.04 | 27.41 ± 1.79 |

image synthesis by a word-level spatial and channel-wise attention-driven generator, a word-level discriminator, and a perceptual loss. DF-GAN (Deep Fusion GAN) [48] proposes a deep text-image fusion block for the generator, and the Matching-Aware Gradient Penalty (MA-GP) for the discriminator. We follow the code released by the authors [64, 20, 49] to train their models on our compositional splits. We use the same training strategy as described in their original papers.

**Evaluation.** We adopt the two commonly used evaluation metrics for text-to-image synthesis, FID and R-Precision, and our newly proposed CLIP-R-Precision metric described in Section 4. We use around 10,000 and 5000 samples to compute FID scores of the C-CUB and C-Flowers test splits, respectively[2]. To compute R-Precision and Clip-R-Precision, we use 11,000 and 7,000 images for C-CUB and C-Flowers datasets, respectively. Moreover, we conduct a human evaluation on Amazon Mechanical Turk to evaluate the quality and correctness of the synthesized images (see Section 5.2).

## 5.1 Automatic Evaluation

We report automatic evaluation scores on the proposed compositional splits of the C-CUB and C-Flowers datasets in Table 2 and 3. We summarize our findings below.

Overall, we observe degradation in most cases in both image quality (indicated by FID) and image correctness (indicated by R-Precision and CLIP-R-Precision) for the unseen and swapped compositions, which verifies our hypothesis that the model cannot generalize well to novel compositions of concepts. In general, degradation is much worse for image correctness (R-Precision and CLIP-R-Precision) than image quality (FID) for both Shape and Color on both datasets, and for both Test Unseen and Test Swapped splits. In other words, the model is able to generate perceptually convincing images, but they do not closely match the input captions.

With a few exceptions, models generally perform worse on the Swapped split than on the Unseen split in C-CUB while the trend is reversed in C-Flowers. We speculate that such difference comes from how much contextual information is inherent in the captions for each dataset. More concretely, even if there are unseen compositions, models may still be able to infer the class/type of the bird or flower based on other descriptions in the caption, thereby allowing the model to generate reasonably correct images without necessarily understanding the novel compositions. In C-CUB, the contextual cues might be stronger which makes Swapped split more confusing while in C-Flowers, the context of the sentence might not be strongly indicative of the flower type, thus showing better performance on the Swapped split.

The trends for different methods (DMGAN, ControlGAN, and DF-GAN) are mostly consistent. For both DMGAN and ControlGAN, the scores of CLIP-R-Precision are significantly lower than

---

[2]We do not report the Inception Score [45], as it is superseded by the FID metric.

Table 3: Benchmarking on the C-Flowers Color and C-Flowers Shape datasets.

| Type | Model | Split | FID↓ | R-Precicion (R=1)↑ | CLIP R-Precision (R=1)↑ |
|------|-------|-------|------|--------------------|--------------------------|
| Color | DMGAN | Seen | 28.76 | 63.42 ± 0.85 | 52.03 ± 2.36 |
| | | Unseen | 31.30 | 48.49 ± 1.83 | 38.46 ± 1.62 |
| | | Swapped | 28.48 | 51.71 ± 2.38 | 44.68 ± 3.15 |
| | ControlGAN | Seen | 36.88 | 49.90 ± 0.87 | 39.28 ± 1.83 |
| | | Unseen | 38.02 | 36.46 ± 0.58 | 26.12 ± 1.24 |
| | | Swapped | 37.35 | 40.32 ± 3.44 | 35.41 ± 3.28 |
| | DF-GAN | Seen | 35.05 | 42.92 ± 1.80 | 40.90 ± 1.79 |
| | | Unseen | 38.24 | 35.63 ± 1.59 | 32.62 ± 1.68 |
| | | Swapped | 39.28 | 34.82 ± 4.02 | 38.31 ± 4.28 |
| Shape | DMGAN | Seen | 28.25 | 61.19 ± 2.10 | 49.07 ± 1.46 |
| | | Unseen | 29.26 | 47.82 ± 1.54 | 34.80 ± 1.78 |
| | | Swapped | 26.64 | 52.05 ± 3.47 | 38.16 ± 1.98 |
| | ControlGAN | Seen | 36.32 | 56.06 ± 1.07 | 33.18 ± 1.57 |
| | | Unseen | 36.99 | 38.10 ± 1.46 | 25.27 ± 1.33 |
| | | Swapped | 35.97 | 40.29 ± 2.81 | 27.47 ± 2.50 |
| | DF-GAN | Seen | 34.04 | 41.52 ± 1.96 | 43.54 ± 1.17 |
| | | Unseen | 35.28 | 33.75 ± 1.59 | 33.60 ± 0.98 |
| | | Swapped | 34.68 | 35.43 ± 2.76 | 34.33 ± 2.40 |

R-Precision, which is because DAMSM is used for both training the models and evaluating the R-Precision. In contrast, DF-GAN does not use DAMSM loss for training, so R-Precision and CLIP-R-Precision results are comparable but lower. Thus, optimizing with DAMSM loss generally improves the retrieval-based evaluation metrics, but makes the model biased towards R-Precision. Our proposed CLIP-R-Precision is a more objective and unbiased metric for evaluating image correctness.

When comparing the Color and Shape splits, the trends are overall similar. On C-CUB, DMGAN and ControlGAN tend to do slightly better on Shape, while DF-GAN does slightly worse. On C-Flowers, the results are mixed. We do not see that either Color or Shape is consistently more challenging.

## 5.2 Human Evaluation

Next we study whether the findings from the automatic metrics are confirmed by a human evaluation. We focus on the Seen and Swapped splits to reduce any additional confounding factors. All images are generated by the DMGAN model, the best performing of the three methods. We separately assess *perceptual quality* and *correctness* of the generated images using Amazon Mechanical Turk.

**Perceptual Evaluation.** We sample 100 caption-image pairs from each of the following splits: C-CUB Color Seen/Swapped, C-CUB Shape Seen/Swapped, C-Flowers Color Seen/Swapped, C-Flowers Shape Seen/Swapped. We design our tasks by pairing images from the Seen and Swapped splits that correspond to an original caption and its swapped variant. (The captions are *not* used during the evaluation.) We ask the workers "Which image looks more realistic?" with the candidate answers being *Image 1*, *Image 2* or *About the same*. For each task we ask 5 workers to provide their judgment. The results are presented in Table 4. We report two metrics: "All" is calculated over all 500 tasks independently[3]; "Majority" is calculated over 100 image pairs, where an image "wins" if at least 3 out of 5 judges agreed on it. Interestingly, we see that for the Color splits the images generated from the Seen captions get higher scores, while the opposite is true for the Shape splits. We posit that the "Swapped" generated images may look realistic despite not being accurate.

**Correctness Evaluation.** Now, we aim to answer the key question in this paper, do the images generated based on the captions with unseen phrases correctly depict the input descriptions? We rely on the same data as described above. Here, each task includes two captions (Seen and Swapped) and one image. An image may be generated either based on a Seen or a Swapped caption. Here, we ask the workers: "Which caption better matches the image?" with the answer choices *Caption 1*, *Caption 2* or *About the same*. The intuition is that if the image was generated correctly (w.r.t. the caption) it should be possible to match it to its caption (and not to a distractor caption). For each task we collect judgments from 5 workers. Table 5 reports the accuracy with which the "Seen" and "Swapped"

---

[3]We have 500 tasks since there are 100 image pairs and 5 judges for each.

Table 4: Perceptual quality evaluation with humans on the C-CUB and C-Flowers dataset. "Maj." stands for Majority vote.

| Type | Model | Split | All | Maj. |
|------|-------|-------|-----|------|
| | | **C-CUB dataset** | | |
| Color | DMGAN | *Seen* is better | **55.8** | **55.0** |
| | | *Swapped* is better | 42.0 | 39.0 |
| Shape | DMGAN | *Seen* is better | 45.2 | 39.0 |
| | | *Swapped* is better | **49.0** | **49.0** |
| | | **C-Flowers dataset** | | |
| Color | DMGAN | *Seen* is better | 48.6 | **50.0** |
| | | *Swapped* is better | **48.8** | 47.0 |
| Shape | DMGAN | *Seen* is better | 41.4 | 40.0 |
| | | *Swapped* is better | **55.0** | **58.0** |

Table 5: Correctness evaluation with humans on the C-CUB and C-Flowers dataset. "Maj." stands for Majority vote.

| Type | Model | Split | All | Maj. |
|------|-------|-------|-----|------|
| | | **C-CUB dataset** | | |
| Color | DMGAN | *Seen* accuracy | **81.6** | **88.0** |
| | | *Swapped* accuracy | 64.2 | 65.0 |
| Shape | DMGAN | *Seen* accuracy | **59.6** | **68.0** |
| | | *Swapped* accuracy | 47.8 | 45.0 |
| | | **C-Flowers dataset** | | |
| Color | DMGAN | *Seen* accuracy | **86.2** | **93.0** |
| | | *Swapped* accuracy | 45.6 | 41.0 |
| Shape | DMGAN | *Seen* accuracy | **67.8** | **71.0** |
| | | *Swapped* accuracy | 44.6 | 39.0 |

images were matched to their corresponding captions. "All" is calculated over 500 tasks ($100 \times 5$) per split; for "Majority" an image accuracy is 1 if at least 3 out of 5 judges matched it to its own caption, and 0 otherwise (100 tasks per split). Here we clearly see that the "Swapped" images are less true to their captions, as evident by significantly lower accuracy than that of the "Seen" images. We note, that even the Shape Swapped samples, which have better perceptual quality according to the human judges, still significantly lag behind the respective Shape Seen samples in terms of image correctness.

Overall, the correctness evaluation clearly shows degradation in quality, while the perceptual evaluation is less telling. Compared to the automatic evaluation results in the previous section, again, the correctness results are consistent, while the perceptual quality results are less so.

**Correlation between Automatic and Human Scores.** Here we assess how well the DAMSM and CLIP embeddings, used to compute R-Precision and CLIP-R-Precision in our automatic evaluation, align with the human scores in the correctness evaluation. Given an image and two captions, each caption gets scored with the ratio $m/5$ if it was selected by $m$ out of 5 human judges. Next, we compute the corresponding scores based on the DAMSM and CLIP embeddings. Finally, we compute Pearson correlation and Spearman correlation between the human scores and DAMSM / CLIP. Pearson correlation is a measure of linear correlation between two sets of data, and Spearman correlation is a measure of monotonic correlation. Our results are presented in Table 6 and Table 7. The Pearson correlation scores in Table 6 are low because the correlation between CLIP/DAMSM similarity scores and human evaluation scores are expected to be monotonic but not necessarily linear. In addition, we also compute the binary decision consensus between human evaluators and the model in Table 8. For each paired seen and swapped caption, and an image generated from either the seen caption or the swapped caption, the binary choice (*i.e.*, "the seen caption matches the image better than the swapped caption" or " the swapped caption matches the image better than seen caption") is made by both human evaluators and the models. We compute the percentage that the human evaluators and the model reach a consensus in the binary decision. We also report the consistency between human evaluators. At each time, we hold out one human evaluator among the five evaluators, and compute the Pearson/Spearman correlation coefficients and the binary decision consensus between the "held out evaluator" and the average scores of the other four evaluators. The five correlation scores, each taking one evaluator as the "held out evaluator", are averaged to obtain the final score. Based on the correlation and consensus analysis, we conclude that the CLIP-based scores have higher correlation with human scores than the DAMSM-based scores. We are aware that both of those evaluation metrics are far from perfect, which is indicated by the relatively low correlation scores. However, our work offers a potential direction to look for an improvement for evaluating the correctness of text-to-image synthesis models.

## 5.3 Qualitative Analysis

In Figure 2, we illustrate the images synthesized by DMGAN given the text inputs containing seen adjective-noun pairs versus their swapped variants (with a fixed latent code). Each row corresponds to the DMGAN model trained on C-CUB Color/Shape or C-Flowers Color/Shape data split. The images synthesized from the Seen captions are in the 1st, 3rd, and 5th columns; for the Swapped captions,

Table 6: Pearson Correlation Coefficient between human eval scores and similarity scores computed from DAMSM and CLIP embeddings. Higher is better.

| Model | C-CUB dataset | | C-Flowers dataset | |
|---|---|---|---|---|
| | Color | Shape | Color | Shape |
| DAMSM | 0.0503 | **0.1229** | -0.0457 | -0.0435 |
| CLIP | **0.0752** | 0.0878 | **0.2818** | **0.0920** |
| Human | 0.5949 | 0.3949 | 0.5891 | 0.3721 |

Table 7: Spearman Correlation Coefficient between human eval scores and similarity scores computed from DAMSM and CLIP embeddings. Higher is better.

| Model | C-CUB dataset | | C-Flowers dataset | |
|---|---|---|---|---|
| | Color | Shape | Color | Shape |
| DAMSM | 0.1224 | 0.0170 | 0.0456 | **0.0876** |
| CLIP | **0.1263** | **0.0806** | **0.3151** | 0.0702 |
| Human | 0.5890 | 0.4007 | 0.5870 | 0.3698 |

Table 8: Binary decision consensus between human eval scores and similarity scores computed from DAMSM and CLIP embeddings. Higher is better.

| Model | C-CUB dataset | | C-Flowers dataset | |
|---|---|---|---|---|
| | Color | Shape | Color | Shape |
| DAMSM | 0.5487 | **0.5269** | 0.5233 | 0.4796 |
| CLIP | **0.5641** | 0.5215 | **0.6528** | **0.5204** |
| Human | 0.8171 | 0.6997 | 0.8048 | 0.6879 |

the images are in the 2nd, 4th, and 6th columns. (More examples are provided in the Supplemental.) In many cases, images generated from swapped captions do not accurately depict their captions. In some cases, an unseen swapped adjective-noun pair in the caption is not only misrepresented in its own corresponding sub-region in the output image, but it also degrades the correctness of other sub-regions. For instance, in the first example from C-CUB Color (top row), "*blue eye*" in the description has results in the color of the "*bill*" and the "*head*" to be "*blue*" as well. This could be due to the entanglements between different attributes or body parts learned by the model, shedding light onto the cause of poor compositional generalization.

To further investigate this issue, we look at a DMGAN model trained on the *original* (non-compositional) training split of the CUB dataset [52]. We generate images from multiple input descriptions where all adjective-noun pairs are seen during training. We then manually swap one or two attributes in the input caption and generate an image from the same latent code, shown in Figure 3. Note that all adjective-noun pairs in the swapped captions are also *seen* during training. Here, we observe an entanglement between the background, body shape, color, pose, etc (*e.g. "long neck"* often results in the image of a *"waterbird"*, *"a white bird with gray wings"* usually appears on a "*beach*", while a "*yellow bird*" sits on a "*branch*"). This reveals that even a non-compositional scenario exhibits entanglement issues. We hypothesize that an efficient disentanglement approach could help address both the entanglement and compositionality of the text-to-image synthesis models.

## 6  Discussion and Broader Impact

In this work, we establish a benchmark for evaluating and analyzing the generalization ability of text-to-image synthesis methods w.r.t. the novel compositions of concepts. We create new data splits for CUB and Oxford-102 datasets (C-CUB and C-Flowers), which are specifically designed for evaluating and diagnosing how well the models can generalize to novel compositions. We also propose a new evaluation metric, named CLIP-R-Precision, which offers a more objective measurement of image correctness. We evaluate three state-of-the-art approaches, DMGAN, ControlGAN, and DF-GAN, on the proposed compositional data splits and evaluation metrics.

One limitation of our work is that we only include three text-to-image synthesis methods in our benchmarking study[4]. Future work should consider more approaches which might lead to more interesting analysis and findings. It is also worth studying other types of compositions besides color and shape, such as novel compositions of objects and novel compositions of object relationships.

This is the first work that establishes a benchmark for compositional text-to-image synthesis. We hope that it will inspire future work on improving compositionality in text-to-image generation. A

---

[4]We have considered including several other methods, but have discovered that in many cases the code was either not released or not complete.

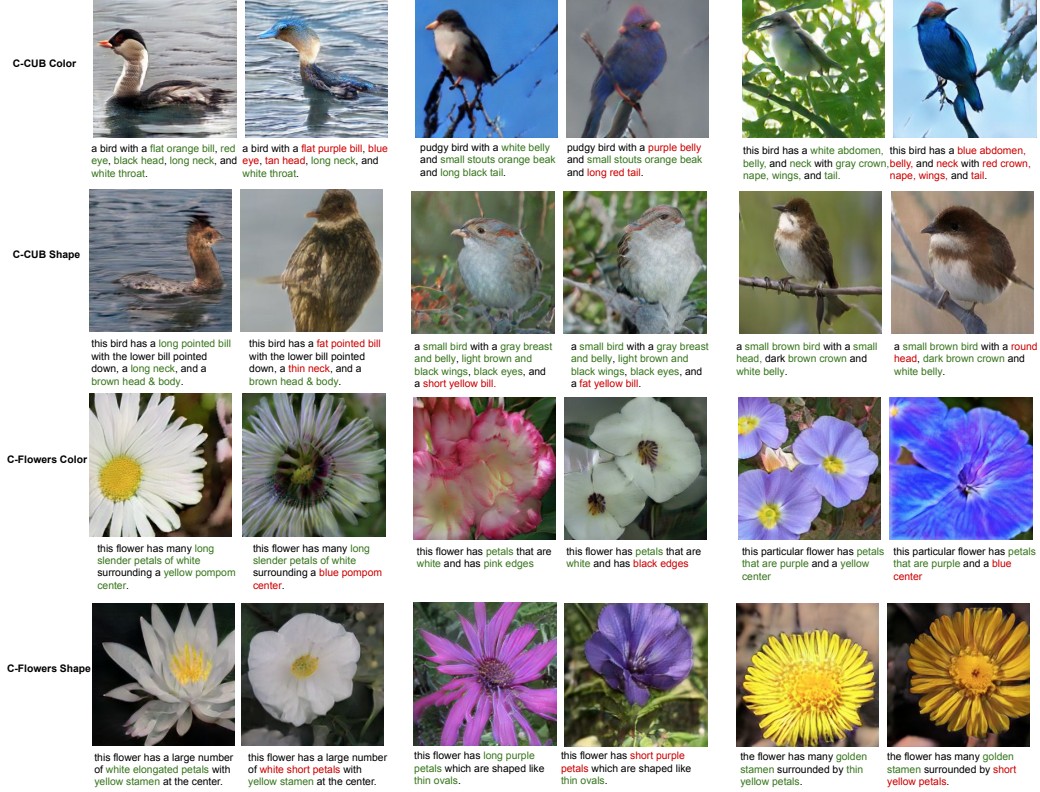

Figure 2: Seen adjective-noun pairs (illustrated in green; 1st, 3rd, and 5th columns) are swapped with unseen adjective-noun pairs (illustrated in red; 2nd, 4th, and 6th columns). A DMGAN model has been trained on our two compositional datasets on the two color and shape splits.

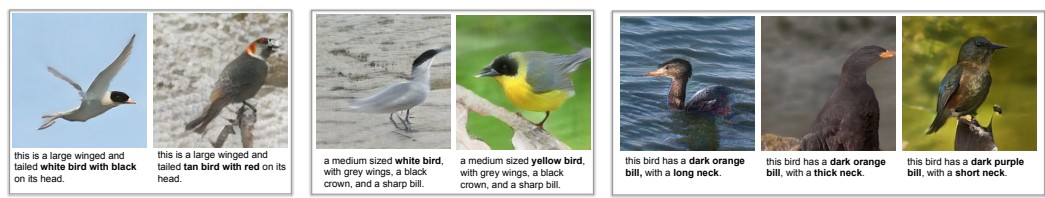

Figure 3: Entanglement between background, shape, color, and pose. All adjective-noun pairs are seen at training time. Images are generated by a DMGAN model trained on the original CUB dataset.

model that performs well on novel compositions would also be more user-controllable and less biased. Such models have the potential to be used for automatic artistic design and image editing. However, image synthesis techniques should be used carefully as they might pose a threat if used for generating fake images that might mislead people. Novel composition of objects, attributes, actions, and scenes while generating an image from an input text could result in fake news in which a fraudulent story matches perfectly with a realistic-looking synthesized image. For instance, one could claim the discovery of an extra-terrestrial on the earth or could attack the reputation of a person by publishing a malicious image of him/her doing an action that never had happened in reality. In this paper, we have studied the novel composition of objects and attributes in the context of birds and flowers, but this study could be extended to other types of composition and other data sets and contexts. We urge the researchers and users to be aware of such consequences. Finally, we use a large-scale web-based CLIP model [38] as part of our metric, which may have encoded some inappropriate biases that could propagate to our metric. We would caution the researchers adopting our CLIP-based metric to more sensitive domains to make sure that no harmful/offensive biases get propagated via the proposed metric.

# 7 Acknowledgement

This work was supported in part by DoD including DARPA's XAI, LwLL, and/or SemaFor programs, as well as BAIR's industrial alliance programs.

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
