# OpenReview forum: "Benchmark for Compositional Text-to-Image Synthesis"
_NeurIPS.cc/2021/Track/Datasets_and_Benchmarks/Round1 — NeurIPS 2021 Datasets and Benchmarks Track (Round 1)_

### Official Review · Reviewer_TuoW · 2021-06-27
**A Systematic Study of the Model Generalization to Novel Compositions in Text-to-Image Synthesis**

**Rating:** 6
**Confidence:** 4
**Clarity:** The paper is generally well-written.

**Strengths:**

- The proposed benchmark is interesting. To the best of my knowledge, it is the first benchmark that systematically studies how well text-to-image synthesis methods generalize to novel compositions of concepts.
- The benchmark may inspire future work on improving compositionality in the text-to-image synthesis field.
- The paper is generally well-written, and the proposed benchmark is explained in detail.
- The experimental results are verified by both automatic metrics and human evaluations. Besides, the qualitative analysis is provided.

**Weaknesses:**

- The benchmark only includes three text-to-image synthesis methods. Since this paper focuses on benchmarking text-to-image methods to study the model generalization to novel compositions of concepts on existing datasets, it would be better to include more baselines to make the analysis more comprehensive and reveal its solid contribution. For instance, it would be more interesting to include state-of-the-art methods, such as DALL-E [1].
- In Table 3, the results of DF-GAN on the C-Flowers dataset are missing.
- The numbers of samples used to compute FID on C-CUB and C-Flowers are $1,100$ and $700$, respectively, which are much less than that in the original paper [2] and state of the arts [3,4] in generative modeling. Would it introduce some potential inaccuracy for the FID evaluation?
- It would be better to include the Seen and Unseen splits for the human evaluation to make the benchmark more comprehensive.


---
[1] Aditya Ramesh, Mikhail Pavlov, et al. Zero-Shot Text-to-Image Generation. arXiv preprint arXiv:2102.12092, 2021.

[2] Martin Heusel, Hubert Ramsauer, et al. GANs Trained by a Two Time-Scale Update Rule Converge to a Local Nash Equilibrium. In NeurIPS, 2017.

[3] Tero Karras, Samuli Laine, et al. Analyzing and Improving the Image Quality of StyleGAN. In CVPR, 2020.

[4] Andrew Brock, Jeff Donahue, et al. Large Scale GAN Training for High Fidelity Natural Image Synthesis. In ICLR, 2018.

**Additional Feedback:**

- Line 9: Typo on the double quotes in Abstract.
- Since each individual concept, *i.e.*, an adjective or a noun, is observed during training, I am uncertain about the clear difference between Test *Unseen* and Test *Swapped*. The descriptions about this part are not clear enough. Are there any concrete examples to help illustrate this?
- What is the exact motivation of Test *Swapped*? The authors mentioned that it is used to prevent other factors besides the novel adjective-noun compositions from affecting the generalization performance. I would like to ask for more intuition about this.

**Correctness:**

The claims made in the submission are correct, verified by empirical studies. The evaluation methods and the experiment design are appropriate in the proposed benchmark.

**Documentation:**

The authors have tried their best to provide details of their benchmark. It would be better to ensure releasing all the necessary source code and training details to support reproducibility of the proposed benchmark.

**Ethics:**

From my perspective, there are few or no ethical concerns that warrant further discussion or review.

**Relation To Prior Work:**

The paper clearly discussed how this work differs from previous contributions.

**Summary And Contributions:**

This paper presents a systematic study of the model generalization to novel text concept compositions for benchmarking representative text-to-image synthesis methods. The authors construct new data splits for the existing Caltech-UCSD Birds and Oxford-102 datasets. The splits are specially designed for evaluating and analyzing how well the model can generalize to unseen compositions of concepts. As a preliminary attempt, they include object-color and object-shape scenarios for the unseen compositions. They also propose a new evaluation metric, named CLIP-R-Precision, based on the multimodal CLIP model, outperforming the commonly used R-Precision.

---

> ### Author Response · Authors · 2021-07-15
> **Review Response**
>
> We would like to thank you for all the constructive feedbacks. We have tried our best in addressing your concerns in the following response:
>
> **TEST SWAPPED VS. TEST UNSEEN**
>
> We would clarify our motivation for Test Swapped through an example as follows.
> Assume a test seen caption is "This is a bird with red beak which is curvy and thick" where "red beak" is a seen composition. While a test unseen caption is "This is a magnificent bird with brown crown, striking yellow cape, orange upper wings, and blue beak". If the quality of the synthesized image from the unseen caption is not good, **it is hard to tell whether it is because of the unseen composition (blue beak) or because the input caption is longer and more complicated**. In order to control the various factors and analyze the effect of novel compositions, we introduce the test swapped caption, "This is a bird with blue beak which is curvy and thick" where it adopts the identical sentence structure and vocabularies with the seen caption, and only the unseen combination of "blue beak" is introduced. This would also **make it easier for the human evaluators to make apples-to-apples comparison of model performance between seen compositions versus the unseen ones**. We have added the above discussion in lines 140-150.
>
> **COMPARE TEST UNSEEN WITH TEST SEEN IN HUMAN EVAL**
>
> Thanks for your suggestion. We will include additional human evaluations in the main paper if our budget and time permit. We would also re-emphasize our motivation for experimenting with Test Swapped in our human evaluation study as discussed above. We aimed at simplifying the problem for the evaluators by comparing two images where their corresponding captions are more comparable to each other (from Test Seen and Test Swapped).
>
>
> **OTHER BASELINES**
>
> We tried training other recent models in text-to-image synthesis but found their code to be either incomplete or not released. We specifically tried DALL-E for which the full model is not available as well as X-lxmert, objGAN, RiFeGAN, CPGAN, and LeicaGAN. We emphasize that we will release our dataset and evaluation code so that other works can evaluate on our benchmark.
>
> **DF-GAN ON THE C-FLOWERS DATASET**
>
> We have trained DF-GAN on the C-Flowers dataset and the evaluation scores are added to Table 3. The results for DF-GAN are consistent with the findings in the main paper (i.e. drop in both image quality and correctness from seen to unseen/swapped splits, etc.).
>
> **NUMBER OF SAMPLES IN COMPUTING FID**
>
> We increased the number of samples used to compute FID to 10,000 and 5,000 for C-CUB and C-Flowers, respectively. The overall FID scores improved across the board; however, the general trend (i.e. test seen scores better than test unseen and test swapped scores) remains the same. We have updated the tables in the main text.

---

> > ### Comment · Reviewer_TuoW · 2021-07-19
> > **Most Concerns were Resolved**
> >
> > Through the authors' response, most of my concerns have been resolved. Overall, I feel positive about this submission.

---

### Official Review · Reviewer_9BfV · 2021-07-02
**Good benchmark but unconvinced by evaluation metric**

**Rating:** 6
**Confidence:** 4

**Strengths:**

Overall, I think the problem space being explored is a worthy area to move into. Text to image generation has seen increased interest over the last few years and having better metrics and benchmarks to evaluate progress is fundamental.

I also found the paper easy to read and understand.


**Weaknesses:**

Why is CLIP model finetuned for the CLIP-R-precision metric trained on only the training splits and not the whole dataset? If the CLIP model is meant to evaluate whether the generated image contains the novel composition, it needs to be able to evaluate the novel composition. If it, itself, hasn’t seen a novel composition, then how will it evaluate the generator? This makes me doubt whether I can trust the CLIP-R-Precision scores reported in the paper. Also, given the low pearson correlation scores (< 0.3) with human evaluation, which is usually categorized as “no correlation” or “very weak” correlation, it further make me doubt the validity of the evaluation metric. This is perhaps the main concern with the paper.

I would also like the paper to report correlation scores between human evaluators. How consistent were people?

There are multiple sentences that are unclear or vague. I mention them in the Clarity section of this evaluation.


**Additional Feedback:**

Paper that should have been cited:

- Sajjadi et al. Assessing Generative Models via Precision and Recall
- Kynkäänniemi et al. Improved Precision and Recall Metric for Assessing
Generative Models
These two papers could have been used to measure that image generation aside from inception score and FID. I suggest adding these papers as citations at the very least. Maybe even talk about whether they can or can not be used to evaluate your conditional generation process.

- Johnson et al. Image Generation from Scene Graphs CVPR 2018
This paper explicitly studies and evaluates whether the generated image contains the concepts in the input scene graph. Ideally, I would have liked to see a comparison of scene graph-to-image generation models and whether the input structure helps make those models better at generalizing. But that might be too large of an ask. So, at least a general discussion on how scene graph to image generation methods compare against text-to-image generation could have been a nice addition.

- Lu et al. Visual Relationship Detection with Language Priors, ECCV 2016
This paper should have been the right citation for relationship detection in line 104 since they introduced the concept of detecting novel compositions of attributes/objects and novel compositions of subject/predicate/object.


**Clarity:**

The “test swapped” split is non-intuitive and there isn’t a clear explanation of why it’s needed or why it was introduced, especially since we already have a “test unseen” split. There is an attempt at an explanation in lines 131 to 135 but it isn’t clear enough. I believe what the paper wanted to convey was that random creating unseen compositions will be testing models on the generation of unrealistic combinations. So, while the “test unseen” split tests for realistic combinations we definitely know exist, the “test swapped” stress tests the model’s compositional generalization. I suggest explaining this a bit better.

Line 140: “We select nouns and adjectives that appear most frequently in the dataset.” Can you be more precise here? Ex, we chose the 100 most frequent object/adjective combinations.

Line 147: “they are 147 randomly sampled from the middle of the sorted list”. Here as well, can you be more specific. If someone had to replicate your dataset creation process, they don’t be able to do it with this sentence. Can you try and add more clarity?

Line 195: “CLIP-R-Precision is a stronger model-agnostic evaluation metric”. I am confused about this. The evaluation metric uses a finetuned CLIP model and so is, by definition, a model-specific evaluation metric. What am I missing here?


**Correctness:**

My main concern is around the evaluation metric. An incorrect evaluation metric will set us back and result in hill climbing on noise. See weakness section.

**Documentation:**

No, please see clarifying section


**Ethics:**

The paper introduces novel splits of an existing dataset and doesn’t release a new dataset in itself. As such, I don’t think there are any new ethics concerns related to the dataset.

That being said, I encourage the authors to write about the general ethical concerns around conditional image generation and the negative consequences of improving generative models that can produce novel compositions.

I also urge the authors to spend less text summarizing the paper in the broader impact section and more on the potential evaluation-related biases that CLIP might introduce.


**Relation To Prior Work:**

Yes

**Summary And Contributions:**

The paper contributes a systematic evaluation of how well text-to-image synthesis models generalize to novel compositions. They introduce two test splits on the CUB dataset and Oxford Flowers dataset. The splits are designed to test whether models can synthesize novel combinations of colors/objects and shape/object combinations. They also introduce a new evaluation metric, called CLIP-R-Precision, to measure how well models generalize to novel compositions and show that it correlates better with human judgment than other metrics. Finally the paper evaluates a few existing text-to-image synthesis models and finds that they perform worse on correctness when given novel compositions.

---

> ### Author Response · Authors · 2021-07-15
> **Review Response**
>
> We would like to thank you for all the constructive feedback. We have tried our best in addressing your concerns in the following response:
>
> **CLIP FINETUNING**
>
>  We train the CLIP model on the entire dataset for evaluation purposes. It has been clarified in lines 210-211 in the paper.
>
> **LOW PEARSON CORRELATION SCORES IN TABLE 6**
>
> We report the consistency of Pearson correlation **between human evaluators** in Table 6 of the main text. At each time, we hold out one human evaluator among the five evaluators, and compute the pearson correlation coefficients between the "held out evaluator" and the average scores of the other four evaluators. The five correlation scores, each taking one evaluator as the "held out evaluator", are averaged to obtain the final score.
> We hypothesize that the Pearson correlation scores in Table 6 are low because its coefficients are initially designed to evaluate linear correlations, while in our case, the correlation between CLIP/DAMSM similarity scores and human evaluation scores are expected to be monotonic but not necessarily linear. We further evaluate the spearman correlation, which measures the monotonic correlation between two variables.
> We also evaluate the **binary decision consensus** between the CLIP/DAMSM similarity scores and the human scores.  For each paired seen and swapped caption, and an image generated from either the seen caption or the swapped caption, the binary choice (i.e., "the seen caption matches the image better than the swapped caption" or "the swapped caption matches the image better than seen caption") is made by both human evaluators and the models. We compute the percentage that the human evaluators and the model reach a consensus in this binary decision.
> **Based on the correlation and consensus analysis, we conclude that CLIP-R-Precision is correlated better with the human scores than R-Precision**. *We are aware that both of those evaluation metrics are far from perfect, which is indicated by the relatively low correlation scores. However, our work offers a potential direction to look for an improvement for evaluating the correctness of text-to-image synthesis models.*
> The above discussions are updated in lines 296-320 in the paper.
>
> **CLIP-R-PRECISION AS A MODEL-AGNOSTIC EVALUATION METRIC**
>
> We state that CLIP-R-Precision is a model-agnostic metric in that the finetuned CLIP **is not involved in model training**. In most prior text-to-image synthesis works, the DAMSM module used for evaluation is also used during training to introduce additional loss/regularization. This results in text-to-image generation systems that are optimized specifically for DAMSM-based evaluation metric (thus, the term model-specific evaluation metric). The issue with such a metric is that models that are not specifically tuned on DAMSM modules will likely perform worse on the metric, thereby necessitating model designs to include DAMSM modules which may not be desirable. When proposing CLIP-R-Precision, our intention was to **disassociate the evaluator (i.e. finetuned CLIP) from text-to-image model training so that the evaluation is as objective as possible and agnostic to the models that are being evaluated**. We have clarified this in the main text in lines 199-215.
>
> **ETHICAL CONCERNS**
>
> Novel composition of objects, attributes, actions, and scenes while generating an image from an input text could result in fake news in which a fraudulent story matches perfectly with a realistic-looking synthesized image. For instance, one could claim the discovery of an extra-terrestrial on the earth or could attack the reputation of a person by publishing a malicious image of him/her doing an action that never had happened in reality. In this paper, we have studied the novel composition of objects and attributes in the context of birds and flowers, but this study could be extended to other types of composition and other data sets and contexts. We urge the researchers and users to be aware of such consequences. We have added the above discussion in lines 361-372.
>
> **DISCUSSION ON EVALUATION-RELATED BIASES INTRODUCED BY CLIP**
>
> Since this section is titled “Discussion and Broader Impact”, we have discussed the key takeaways from the paper before studying its broader impacts. We have mentioned that CLIP may have encoded some biases/stereotypes (e.g. about gender or race), as discussed by the original authors. Since we are not working with human data, we have not studied in depth the effect this may have on our metric. We would caution the researchers adopting our CLIP-based metric to more sensitive domains to make sure that no harmful/offensive biases get propagated via the proposed metric. We have added the above discussion in lines 368-372.
>
> continued..

---

> > ### Author Response · Authors · 2021-07-15
> > **Review Response**
> >
> > **INTUITION BEHIND TEST SWAPPED**
> >
> > We would clarify our motivation for Test Swapped through an example as follows.
> > Assume a test seen caption is "This is a bird with red beak which is curvy and thick" where "red beak" is a seen composition. While a test unseen caption is "This is a magnificent bird with brown crown, striking yellow cape, orange upper wings, and blue beak". If the quality of the synthesized image from the unseen caption is not good, **it is hard to tell whether it is because of the unseen composition (blue beak) or because the input caption is longer and more complicated**. In order to control the various factors and analyze the effect of novel compositions, we introduce the test swapped caption, "This is a bird with blue beak which is curvy and thick" where it adopts the identical sentence structure and vocabularies with the seen caption, and only the unseen combination of "blue beak" is introduced. This would also **make it easier for the human evaluators to make apples-to-apples comparison of model performance between seen compositions versus the unseen ones**. We have added the above discussion in lines 140-150.
> >
> > **DATASET CREATION: SELECTION OF FREQUENT OBJ/ADJ COMBINATIONS**
> >
> > When selecting adjectives, we first determine the 60 most frequent adjectives that appear in the dataset. Then we filter out ones that are either color-related or shape-related. For the C-CUB dataset, there are 11 color-related and 15 shape-related adjectives. For the C-Flower dataset, there are 10 color-related and 12 shape-related adjectives. Then, we find 100 most frequent adjective-object pairs that are associated with the selected adjectives and extract all the nouns. As a result, there are 25 and 19 nouns for the C-CUB and C-Flower datasets, respectively. Given the set of nouns and adjectives, we select the “novel” adjective-noun pairs that will constitute the held-out set. We have included such details and clarifications in the main text (L152-166).
> >
> > **DATASET CREATION: RANDOM SAMPLING FROM THE LIST**
> >
> > We appreciate the feedback. The heldout pairs are randomly sampled from between the 25th and 75th percentiles of the sorted list. We have added more details and clarity regarding the dataset generation process in the main text (L152-166). We will **also release the code to generate the compositional splits for better reproducibility**.
> >
> > **CITATION: PRECISION AND RECALL**
> >
> >  Precision and recall can be used as alternative metrics to FID and IS to measure the quality and diversity of the synthesized samples both in the conditional and unconditional image generation problems. However, all these metrics ignore the correspondence between the input text and the generated image in the conditional text-to-image problem, which is usually measured by R-precision instead. We have followed the literature in text-to-image synthesis and reported the FID scores for different models to measure the quality of samples irrespective of their inputs and use R-precision as well as our proposed CLIP R-Precision to measure the correctness of the synthesized images given their input text. A discussion has been added to the related work section in lines 96-104.
> >
> > **CITATION: SCENE GRAPH TO IMAGE AND VISUAL RELATIONSHIP DETECTION**
> >
> > Thanks for your suggestion. We have added a discussion in the related works section (L90-93, L110-111).

---

### Official Review · Reviewer_cGaD · 2021-07-04
**An adequate systematic study of text-to-image generation methods.**

**Rating:** 7
**Confidence:** 4
**Correctness:** The problem definition and methodolog…
**Clarity:** The paper is well written and easy to…

**Strengths:**

- Overall a well written and clear paper. The authors state clearly what they aim to do, and this paper is easy to follow. The problem description is well done, as are description of experiments. These make the accessibility and accountability of this paper high. I believe the broader research community will be interested in the benchmarking results, although may not be interested in the new evaluation metric.

**Weaknesses:**

- CLIP-R-Precision seems like an incremental evaluation metric, as it replaces the DAMSM in R-Precision with CLIP instead. It would have seem more compelling to try several models, rather than only CLIP, and demonstrate the deficiencies of R-Precision more robustly.

- Minimal discussion of the ethical or societal implications.

**Additional Feedback:**

- Table 6 is a bit surprising, given how low all the correlation are regardless of model. Can the authors comment?

-

**Documentation:**

No code is provided, with a promise for release later on.
No hosting, licensing or maintenance plan or discussion on how new models could be submitted for comparison.
Adequate detail in the supplemental to support reproducibility.

**Ethics:**

Minimal ethics/responsible use discussion. While the benchmarking focused on birds and flowers data, the scenarios themselves are not necessarily only pertinent to birds and flowers. I challenge the authors to discuss ethical impacts in other contexts.

**Relation To Prior Work:**

- The related works focus on new GAN models, rather than on comparable benchmarking for visual composition. I am also curious regarding other metrics besides IS, FID, R-Precision.

**Summary And Contributions:**

This work presents benchmarking of two compositional scenarios (object-colour, object-shape) using birds and flowers datasets. The authors also incorporate a new evaluation metric that is meant to avoid model bias. They explore the scenarios, and their new metric, using three different text-to-image synthesis methods.

---

> ### Author Response · Authors · 2021-07-15
> **Review Response**
>
> We would like to thank you for all the constructive feedback. We have tried our best in addressing your concerns in the following response:
>
> **CONCERNS ABOUT CLIP-R-PRECISION**
>
> The main motivation behind proposing CLIP-R-Precision is to promote an evaluation system that is **not directly involved in the text-to-image model training**. *Which* specific model is to be used to compute R-precision is not as important as *how* the metric should be computed. For this reason, and given the limited time and resources, we decided to use CLIP which was one of the most powerful vision-language joint embedding models at the time. We had some initial exploration on using attribute classifiers as a measure of correctness, yet the lack of fine-grained attribute labels in the Oxford Flowers dataset made further exploration difficult.
>
> **ETHICAL OR SOCIETAL IMPLICATIONS**
>
> Novel composition of objects, attributes, actions, and scenes while generating an image from an input text could result in fake news in which a fraudulent story matches perfectly with a realistic-looking synthesized image. For instance, one could claim the discovery of an extra-terrestrial on the earth or could attack the reputation of a person by publishing a malicious image of him/her doing an action that never had happened in reality. In this paper, we have studied the novel composition of objects and attributes in the context of birds and flowers, but this study could be extended to other types of composition and other data sets and contexts. We urge the researchers and users to be aware of such consequences. We have added the above discussion in lines 361-372.
>
> **LOW CORRELATION SCORES IN TABLE 6**
>
> We report the consistency of Pearson correlation **between human evaluators** in Table 6 of the main text. At each time, we hold out one human evaluator among the five evaluators, and compute the pearson correlation coefficients between the "held out evaluator" and the average scores of the other four evaluators. The five correlation scores, each taking one evaluator as the "held out evaluator", are averaged to obtain the final score.
> We hypothesize that the Pearson correlation scores in Table 6 are low because its coefficients are initially designed to evaluate linear correlations, while in our case, the correlation between CLIP/DAMSM similarity scores and human evaluation scores are expected to be monotonic but not necessarily linear. We further evaluate the spearman correlation, which measures the monotonic correlation between two variables.
> We also evaluate the binary decision consensus between the CLIP/DAMSM similarity scores and the human scores.  For each paired seen and swapped caption, and an image generated from either the seen caption or the swapped caption, the binary choice (i.e., "the seen caption matches the image better than the swapped caption" or "the swapped caption matches the image better than seen caption") is made by both human evaluators and the models. We compute the percentage that the human evaluators and the model reach a consensus in this binary decision.
> Based on the correlation and consensus analysis, we conclude that CLIP-R-Precision is correlated better with the human scores than R-Precision. **We are aware that both of those evaluation metrics are far from perfect, which is indicated by the relatively low correlation scores. However, our work offers a potential direction to look for an improvement for evaluating the correctness of text-to-image synthesis models.**
> The above discussions are updated in lines 296-320 in the paper.
>
> **RELATED METRICS FOR BENCHMARKING VISUAL COMPOSITION**
>
> Precision and recall can be used as alternative metrics to FID and IS to measure the quality and diversity of the synthesized samples both in the conditional and unconditional image generation problems. However, all these metrics ignore the correspondence between the input text and the generated image in the conditional text-to-image problem, which is usually measured by R-precision instead. We have followed the literature in text-to-image synthesis and reported the FID scores for different models to measure the quality of samples irrespective of their inputs and use R-precision as well as our proposed CLIP R-Precision to measure the correctness of the synthesized images given their input text. A discussion has been added to the related work section. The above discussions are updated in lines 96-104 in the main text.
>
> **DOCUMENTATION**
>
> The code for training the models in our experiments are all publicly available. The code for dataset split generation and evaluation will be released shortly. There are no hosting, licensing or maintenance issues because we are proposing a new benchmark/task based on already existing datasets.

---

> > ### Comment · Reviewer_cGaD · 2021-07-19
> > **No further concerns.**
> >
> > Thank you for the detailed response. I am more positive about this submission and have bumped up my score.

---

### Decision · Program_Chairs · 2021-07-26

**Decision:**

Accept

**Comment:**

The paper presents a benchmark for measuring compositionality in text-to-image synthesis. Reviewers found the benchmark novel and well-motivated, and felt the paper was clear and easy to follow. Reviewers pointed out some minor weaknesses in the paper (confusion about low Pearson correlations in Table 3, not enough discussion of ethical implications, confusion about test-swapped split, missing results for DF-GAN on C-Flowers, number of samples used to compute FID) but the author responses and revised paper were able to satisfactorily resolve these issues. Congratulations on having your paper accepted to the NeurIPS 2021 Track on Datasets and Benchmarks! The authors are encouraged to take the feedback from reviewers into account when preparing the camera-ready version of the paper.